# Weisfeiler and Leman follow the Arrow of Time: Expressive Power of Message Passing in Temporal Event Graphs

## Abstract

An important characteristic of temporal graphs is how the directed arrow of time influences their *causal topology*, i.e. which nodes can possibly influence each other causally via time-respecting paths. The resulting patterns are often neglected by temporal graph neural networks (TGNNs). To formally analyze the expressive power of TGNNs, we lack a generalization of graph isomorphism to temporal graphs that fully captures their causal topology. Addressing this gap, we introduce the notion of *consistent event graph isomorphism*, which utilizes a time-unfolded representation of time-respecting paths in temporal graphs. We compare this definition with existing notions of temporal graph isomorphisms. We illustrate and highlight the advantages of our approach and develop a temporal generalization of the Weisfeiler-Leman algorithm to heuristically distinguish non-isomorphic temporal graphs. Building on this theoretical foundation, we derive a novel message passing scheme for temporal graph neural networks that operates on the event graph representation of temporal graphs. An experimental evaluation shows that our approach performs well in a temporal graph classification experiment.

## 1 Motivation

Graph neural networks (GNNs) have become a cornerstone of deep learning in relational data. They have recently been generalized to temporal GNNs (TGNNs) that capture patterns in time series data on *temporal graphs*, where edges carry timestamps. Such temporal graphs are often categorized into two different types: In *discrete-time temporal graphs* (DTTGs), edges carry coarse-grained timestamps, e.g. yearly snapshots with many edges having identical timestamps. Such temporal graphs can naturally be represented as sequences of static snapshot, where each snapshot includes all edges occurring at a given timestamp. The resulting sequence of snapshots naturally lends itself to a generalization of static network analysis or graph learning techniques, as each individual snapshot can be interpreted as a static graph. In contrast, in *continuous-time temporal graphs* edges carry high-resolution, possibly unique timestamps. This implies that snapshot graphs are very sparse, which requires (i) a coarse-graining of time that –however– destroys important temporal information, or (ii) learning techniques able to utilize full temporal information.

To address end-to-end learning tasks in temporal graphs, different TGNN architectures have been proposed, which –depending on the architecture– capture different patterns in temporal graphs. Examples include the evolution of node embeddings in consecutive snapshots for discrete-time temporal graphs, or temporal edge activation patterns in continuous-time temporal graphs Longa et al. (2023). An important additional characteristic of temporal graphs is how the directed *arrow of time* influences their *causal topology*, i.e., which nodes can possibly influence each other causally via time-respecting paths. As example, consider a temporal graph with two edges connecting *Alice* to *Bob* at timestamp $t$ and *Bob* to *Carol* at timestamp $t'$. If $t < t'$, Alice may possibly (but not necessarily) causally influence Carol via Bob. Conversely, if $t' < t$, a causal influence from Alice to Carol is impossible because it would have to propagate backwards in time. To prevent wrong interpretations of the term *causal* in our work, we stress that the correct temporal order of edges is a necessary but not a sufficient condition for causal influence. Hence, considering the arrow of time (and thus the temporal ordering of events) is an important precondition that enables causality-aware learning in temporal graphs.

Numerous works, e.g. in network science, have studied how the temporal ordering of edges in continuous-time temporal graphs influences connectivity, dynamical processes like spreading or diffusion, node centralities, cluster patterns, or controllability Lentz et al. (2013); Rosvall et al. (2014); Scholtes et al. (2014; 2016); Badie-Modiri et al. (2022); Pfitzner et al. (2013). These patterns are often neglected by TGNNs, which can limit their performance in high-resolution time series data on temporal graphs. To formally analyze this aspect in existing TGNNs, in line with works on the expressivity of (static) GNNs Xu et al. (2019); Morris et al. (2019), we lack a generalization of graph isomorphism to temporal graphs that captures how their *causal topology* is shaped by the arrow of time. This could inform the development of new *causality-preserving* message passing schemes for TGNNs with provable expressive power. Addressing this gap, our contributions are:

- We propose a new temporal generalization of graph isomorphism called *time-respecting path isomorphism*, which focuses on the preservation of time-respecting paths between temporal graphs. While it can also be applied to snapshot-based temporal graphs, our definition is specifically suitable for temporal graphs, where edges exhibit high-resolution, possibly unique timestamps.
- An important feature of our definition is that it preserves temporal reachability: if there is a time-respecting path from $u$ to $v$ in $G_1$, there must also be a time-respecting path between the corresponding nodes $u'$ and $v'$ in any $G_2$ isomorphic to $G_1$. The causal structure of a temporal graph implies a partial ordering of timestamped edges: if there is a time-respecting path that includes edge $e$ before edge $e'$, then $e$ must occur before $e'$. However, if no time-respecting path includes both $e$ and $e'$, then it does not matter which one comes first. According to our definition, two temporal graphs can be isomorphic even if the total orderings of timestamped edges are different, as long as the partial ordering imposed by time-respecting paths is the same.
- We contrast our definition with the recently proposed *timewise isomorphism*, which generalizes graph isomorphism to snapshots of discrete-time temporal graphs Walega & Rawson (2025). We show that our definition is less strict. In particular, two temporal graphs may be timewise non-isomorphic even if both exhibit the same time-respecting paths and temporal reachability.
- We show that time-respecting path isomorphism is equivalent to static graph isomorphism on the *augmented event graph*, an auxiliary graph that (i) captures time-respecting paths in the temporal graph through a static line graph expansion, and (ii) is augmented by the original nodes in the temporal graph. This allows us to propose a generalization of the Weisfeiler-Leman (WL) algorithm, which heuristically distinguishes non-isomorphic graphs, to the temporal setting.
- We use our insights to derive a novel message passing scheme operating on the augmented event graph, which generates representations that allow to distinguish non-isomorphic temporal graphs. We show that this has the same expressive power as the WL test on the augmented event graph. We experimentally evaluate the TGNN architecture that follows from our theoretical insights in a temporal graph classification task with synthetic and real datasets.

Our work contributes to the theoretical foundation of temporal graph learning, providing a basis for the development and investigation of neural message passing architectures that consider how the arrow of time shapes the causal topology in temporal graphs.

## 2 RELATED WORK

Over the past years, a number of works have introduced various generalizations of GNNs for temporal data. Following the taxonomy given in Longa et al. (2023), these works can be broadly categorized into snapshot- and event-based models. *Snapshot-based models* operate on data with low temporal resolution that provide a sequence of static graphs. In contrast, *event-based models* operate on high-resolution time series data that capture individual events like the addition or removal of nodes or edges. Here we briefly summarize architectures for these different approaches.

The snapshot-based dynamic graph learning framework ROLAND (You et al., 2022) uses recurrent neural networks (RNNs) to model the evolution of node embeddings generated by applying static graph representation learning methods to a sequence of snapshots. EvolveGCN (Pareja et al., 2020) uses a RNN or LSTM to generate evolving parameters of a graph convolutional network based on a sequence of snapshots. Taking an event-based perspective, the temporal graph network (TGN) architecture (Rossi et al., 2020) integrates embedding and memory modules to capture multi-faceted patterns in sequences of timestamped edges. TGAT (Xu et al., 2020) extends the GAT attention mechanism(Veličković et al. (2018)) to obtain temporal encodings based on the changing neighbor-

hood of nodes in temporal graphs. Wang et al. (2021) introduces Causal Anonymous Walks (CAWs), which are temporal random walks anonymized by node hitting counts, enabling inductive representation learning of temporal networks by capturing causal motifs without relying on node identities or rich edge attributes. None of these methods explicitly model patterns that are due to how the *arrow of time* influences time-respecting paths in temporal graphs. A number of works in network science investigated this aspect in temporal graphs (Holme, 2015). Several works (Lentz et al., 2013; Pfitzner et al., 2013; Oettershagen et al., 2020; Oettershagen & Mutzel, 2022; Badie-Modiri et al., 2022) consider how correlations in the temporal ordering of edges influence connected components, epidemic spreading, and percolation in temporal graphs. Rosvall et al. (2014) study how the temporal ordering of nodes along time-respecting flows influence spreading processes, node centralities and cluster patterns. Scholtes et al. (2014) use a higher-order Laplacian to analytically predict how the temporal ordering of edges speeds up or slows down diffusion. Building on this idea, Scholtes (2017) use higher-order De Bruijn graphs to model time-respecting paths in temporal graphs. Applying this idea to deep learning, Qarkaxhija et al. (2022) generalize neural message passing to higher-order De Bruijn graphs, obtaining a TGNN that models how the arrow of time influences time-respecting paths. Oettershagen et al. (2020) suggest a transformation into a static line-graph for which they propose a graph kernel based on the Weisfeiler-Leman algorithm to classify dissemination processes in networks. We study the connections of this model to our work in Section 4.

In recent years, there has been growing interest in understanding the expressive power of GNN models at a theoretical level. Broadly speaking, expressivity in the context of a GNN architecture refers to its capacity to capture complex structures and distinguish between different graphs. In this paper, our measure of expressive power is the ability of a GNN to produce distinct representations for non-isomorphic graphs. A key insight from recent theoretical work is that the expressive power of message-passing GNNs is fundamentally limited by the 1-dimensional Weisfeiler-Leman (1-WL) graph isomorphism test (Xu et al., 2019; Morris et al., 2019). Moreover, both works independently showed that there exist GNNs that are as least as powerful as the 1-WL test (e.g., the Graph Isomorphism Network by Xu et al. (2019)). These insights have led to a flurry of research on pushing GNN expressivity beyond the 1-WL barrier. One line of work builds on more powerful extensions of the 1-WL test, such as the $k$-dimensional WL test (Morris et al., 2020) or a variant that incorporates edge directions (Rossi et al., 2023). Another line of work augments message passing with additional structural information beyond the raw graph connectivity (e.g., Bouritsas et al. (2023)), or use unique node identifiers (Vignac et al., 2020) or random features (Abboud et al., 2021) in order to break symmetries. While these tricks can increase expressivity, they can also introduce challenges such as overfitting or reliance on problem-specific features. In fact, Franks et al. (2024) show that higher expressivity does not always imply better generalization performance. For a detailed review on WL-based approaches and potential future directions, see, e.g., Morris et al. (2024; 2023).

Compared to the static setting, the expressive power of temporal GNNs has not been explored as thoroughly. This is in part because the time dimension introduces an additional degree of freedom, and thus there is no universally agreed-upon definition of temporal graph isomorphism. Beddar-Wiesing et al. (2024) propose a notion of isomorphism for dynamic graphs, which can be seen as snapshot-based temporal graphs, which all snapshots are considered independently of each other. Walega & Rawson (2025) observe that this does not fully capture the expressive power of two important classes of TGNN architectures: global and local. Instead, they propose *timewise isomorphism*, which enforces consistency between the snapshots. They show that global and local TGNNs differ in their abilities to detect timewise isomorphism, and neither is strictly more powerful than the other. Similarly, Gao & Ribeiro (2022) interpret a temporal graph as a static multi-relational graph in which the timestamps are edge attributes. Hence, an isomorphism must preserve their exact values. Souza et al. (2022) use this notion of isomorphism to propose a temporal generalization of the 1-WL test and show that it has the same expressive power as the message passing architectures TGN, TGAT, and CAW introduced in Rossi et al. (2020); Wang et al. (2021); Xu et al. (2020). They then introduce PINT, a provably more expressive temporal GNN that combines injective temporal message passing with relative positional features that encode how nodes relate in time.

To the best of our knowledge, no existing notions of temporal graph isomorphism precisely capture the influence of time-respecting reachability, which is crucial to understand the evolution of dynamical processes in temporal graphs. In particular, notions like timewise isomorphism are too strict for our purpose as they require exact values of timestamps to be preserved, even if this has no influence on the existence of time-respecting paths.

## 3 PRELIMINARIES

A directed, labeled (static) graph $G = (V, E, \ell_V, \ell_E)$ consists of a set $V$ of nodes, a set $E \subseteq V \times V$ of directed edges, a *node labeling* $\ell_V \colon V \to \mathcal{L}_V$ and an *edge labeling* $\ell_E \colon E \to \mathcal{L}_E$, with countable sets $\mathcal{L}_V$ and $\mathcal{L}_E$. In unlabeled graphs, we omit $\ell_V$ or $\ell_E$ accordingly. For a node $v$, we denote its incoming neighbors by $N_I(v) = \{u \mid (u, v) \in E\}$ and its outgoing neighbors by $N_O(v) = \{u \mid (v, u) \in E\}$. Finally, we define the set of paths $P(G)$ as the set of all alternating node/edge sequences $(v_0, e_1, v_1, e_2, \ldots, e_k, v_k)$ with $e_i = (v_{i-1}, v_i) \in E$ for $i \in \{1, \ldots, k\}$. Note that we do not distinguish between walks and paths or, equivalently, do not require paths to be simple.

**Definition 1** (Graph isomorphism). *For two static graphs $G_1 = (V_1, E_1, \ell_V^1, \ell_E^1)$ and $G_2 = (V_2, E_2, \ell_V^2, \ell_E^2)$, an* isomorphism *is a bijective mapping $\pi \colon V_1 \to V_2$ with these properties:*

*(i) Edge-preserving:* $\quad (u, v) \in E_1 \iff (\pi(u), \pi(v)) \in E_2 \quad \forall u, v \in V$
*(ii) Node label-preserving:* $\ell_V(u) = \ell_V(\pi(u)) \qquad\qquad\qquad\qquad \forall u \in V$
*(iii) Edge label-preserving:* $\ell_E(u, v) = \ell_E(\pi(u), \pi(v)) \qquad\qquad \forall (u, v) \in E$

*We say that the graphs $G_1$ and $G_2$ are* isomorphic *iff such a mapping $\pi$ exists.*

**Definition 2** (Temporal graph). *We define a (directed) temporal graph as $G^\tau = (V, E^\tau)$, where $V$ is the set of nodes and $E^\tau \subseteq V \times V \times \mathbb{N}$ is the set of timestamped edges, i.e., an edge $(u, v; t) \in E^\tau$ describes an interaction between $u$ and $v$ at time $t$.*

Note that timestamped edges represent instantaneous events, i.e., $(u, v; t) \in E^\tau$ does not imply $(u, v; t') \in E^\tau$ for $t \neq t'$. Like Oettershagen et al. (2020), we assume a unit edge traversal time. Following Pan & Saramäki (2011), we assume a maximum time difference $\delta$ between consecutive edges in the following definition of time-respecting paths Pan & Saramäki (2011). This is crucial as we often consider temporal graphs where the observation period is much longer than the timescale of processes of interest. As an example, in a social network with timestamped interactions observed over multiple years, information typically propagates within hours or days, i.e. we are not interested in paths where consecutive edges occur in different years.

**Definition 3** (Time-respecting path). *A path of length $k$ in a temporal graph $G^\tau = (V, E^\tau)$ is an alternating sequence of nodes and timestamped edges $p = (v_0, e_1, v_1, \ldots, e_k, v_k)$ with $e_i = (v_{i-1}, v_i; t_i) \in E^\tau$ for $i \in \{1, \ldots, k\}$. For a maximum time difference (or waiting time) $\delta \in \mathbb{N}$, we say that $p$ is* time-respecting *if $1 \leq t_i - t_{i-1} \leq \delta$ for $i \in \{1, \ldots, k\}$. We denote the set of time-respecting paths in $G^\tau$ as $P^\tau(G^\tau)$.*

The structure of time-respecting paths can be encoded in the temporal event graph, which is a static graph whose nodes are the timestamped edges. Two nodes are connected by an edge if the corresponding timestamped edges form a time-respecting path of length two.

**Definition 4** (Temporal event graph). *Let $G^\tau = (V, E^\tau)$ be a temporal graph with waiting time $\delta$. The temporal event graph is given by $G^\mathcal{E} = (E^\tau, \mathcal{E})$ with*

$$\mathcal{E} = \{((u, v; t), (v, w; t')) \mid (u, v; t), (v, w; t') \in E^\tau, 1 \leq t' - t \leq \delta\}.$$

Note that the time-respecting paths of length $k \geq 2$ in $G^\tau$ correspond to the paths of length $k - 1$ in $G^\mathcal{E}$, whereas the time-respecting paths of length 1 in $G^\tau$ correspond to the nodes in $G^\mathcal{E}$.

Furthermore, we consider two static representations of temporal graphs.

**Definition 5** (Time-aggregated/concatenated static graph). *For a temporal graph $G^\tau = (V, E^\tau)$, let $t_{min}(G^\tau) = \min\{t \mid (u, v; t) \in E^\tau\}$ denote the earliest timestamp that occurs in $G^\tau$. For each pair of nodes $u, v \in V$, let $T(u, v) = \{t - t_{min}(G^\tau) \mid (u, v; t) \in E^\tau\}$ denote the set of timestamps at which the edge $(u, v)$ occurs, relative to the earliest timestamp. Then the set of* static edges *is given by $E^s = \{(u, v) \mid T(u, v) \neq \emptyset\}$. The* time-aggregated static graph *of $G^\tau$ is the directed, edge-labeled graph $G^a = (V, E^s, \ell^a)$ with edge labels $\ell^a(u, v) = |T(u, v)|$. The* time-concatenated static graph *of $G^\tau$ is the directed, edge-labeled graph $G^c = (V, E^s, \ell^c)$ with edge labels $\ell^c(u, v) = T(u, v)$.*

The time-aggregated static graph $G^a$ is a lossy representation of a temporal graph that (i) preserves the topology and frequency of timestamped edges, but (ii) discards information on the time. The existence of a time-respecting path in $G^\tau$ implies the existence of a corresponding path in $G^a$.

However, the converse is not true: a path between nodes $u$ and $v$ in $G^a$ may exist even if there is no time-respecting path between $u$ and $v$ in $G^a$. By contrast, the time-concatenated static graph labels each static edge with the set of timestamps at which the edge exists, so the representation is lossless.

## 4    ISOMORPHISMS IN TEMPORAL GRAPHS

To motivate our temporal generalization of graph isomorphism, we make the following observation.

**Observation 1.** *Let $\pi\colon V_1 \to V_2$ be a bijective node mapping between two graphs $G_1 = (V_1, E_1)$ and $G_2 = (V_2, E_2)$. For any edge $e = (u, v) \in E$, we write $\pi(e) = (\pi(u), \pi(v))$. Then $\pi$ is edge-preserving if and only if it is* path-preserving, *i.e., the following holds for all alternating node/edge sequences $(v_0, e_1, v_1, \ldots, e_{k-1}, v_k)$ with $k \in \mathbb{N}$:*

$$(v_0, e_1, v_1, \ldots, e_{k-1}, v_k) \in P(G_1) \iff (\pi(v_0), \pi(e_1), \pi(v_1), \ldots, \pi(e_{k-1}), \pi(v_k)) \in P(G_2).$$

This is due to the fact that adjacent edges transitively expand into paths. Thus, two isomorphic static graphs are topologically equivalent in terms of edges *and* paths. Importantly, this property does not directly translate to time-respecting paths in temporal graphs: two adjacent timestamped edges $(u, v; t)$ and $(v, w; t')$ only form a time-respecting path if $1 \le t' - t \le \delta$. Hence, a temporal generalization of graph isomorphism should preserve not only the timestamped edges, but also the *causal topology* in terms of time-respecting paths. Conversely, we are interested in an isomorphism definition that does not force the *values* of timestamps to be preserved, provided that the resulting time-respecting paths in two temporal graphs are identical.

**Definition 6** (Time-respecting path isomorphism). *Let $G_1^\tau = (V_1, E_1^\tau)$ and $G_2^\tau = (V_2, E_2^\tau)$ be two temporal graphs. We say that $G_1^\tau$ and $G_2^\tau$ are* time-respecting path isomorphic *if there is a bijective node mapping $\pi_V\colon V_1 \to V_2$ and a bijective timestamped edge mapping $\pi_E\colon E_1^\tau \to E_2^\tau$ such that the following holds for all alternating node/edge sequences $(v_0, e_1, v_1, \ldots, e_{k-1}, v_k)$ with $k \in \mathbb{N}$:*

$$(v_0, e_1, v_1, \ldots, e_{k-1}, v_k) \in P^\tau(G_1^\tau)$$
$$\iff (\pi_V(v_0), \pi_E(e_1), \pi_V(v_1), \ldots, \pi_E(e_{k-1}), \pi_V(v_k)) \in P^\tau(G_2^\tau).$$

Note that in contrast to Observation 1, this definition also includes an edge mapping. In a static graph, each edge is uniquely defined by a pair of endpoints, so the edge mapping is induced by the node mapping. This is not the case in temporal graphs, in which multiple timestamped edges may connect the same node pair at different times. Hence, the edge mapping is specified separately.

A drawback of this isomorphism definition is that it appears difficult to test, since the number of time-respecting paths may be exponential in the graph size. Therefore, we derive equivalent notions of temporal graph isomorphism that are easier to test. In order to preserve paths of length 1, which consist of a single timestamped edge $e = (u, v; t)$ and are always time-respecting, we must ensure that $\pi_E(e)$ connects $\pi_V(u)$ to $\pi_V(v)$. We call this property *node consistency*. Node-consistent mappings preserve paths, but not necessarily their time-respecting property. To ensure this, we observe that time-respecting paths of length $k \ge 2$ correspond to paths in the temporal event graph. We can preserve them by requiring $\pi_E$ to be path-preserving between the temporal event graphs.

**Definition 7** (Consistent event graph isomorphism). *Let $G_1^\tau = (V_1, E_1^\tau)$ and $G_2^\tau = (V_2, E_2^\tau)$ be two temporal graphs with corresponding temporal event graphs $G_1^\mathcal{E} = (E_1^\tau, \mathcal{E}_1)$ and $G_2^\mathcal{E} = (E_2^\tau, \mathcal{E}_2)$. A mapping $\pi_E\colon E_1^\tau \to E_2^\tau$ is a* consistent event graph isomorphism *if and only if*

   (i) *there exists a mapping $\pi_V\colon V_1 \to V_2$ such that*
$$\forall (u, v; t) \in E_1^\tau \quad \exists t'\colon \pi_E(u, v; t) = (\pi_V(u), \pi_V(v); t'), \text{ and}$$
   (ii) *$\pi_E$ is a graph isomorphism between $G_1^\mathcal{E}$ and $G_2^\mathcal{E}$.*

This definition can be simplified further by constructing an *augmented event graph* (see Figure 1), which encodes the node consistency property in its topology. In this way, we reduce the problem of testing for time-respecting path isomorphism to the problem of testing for static graph isomorphism on the augmented event graphs.

**Definition 8** (Augmented event graph). *Let $G^\tau = (V, E^\tau)$ be a temporal graph with event graph $G^\mathcal{E} = (E^\tau, \mathcal{E})$. The* augmented event graph *is the static, directed, node-labeled graph $G^{aug} = (V^{aug}, E^{aug}, \ell)$ with $V^{aug} = V \cup E^\tau$, $E^{aug} = \mathcal{E} \cup E^{out} \cup E^{in}$ and*

$$\ell(v) = \begin{cases} 0 & \text{if } v \in V, \\ 1 & \text{if } v \in E^\tau, \end{cases} \qquad \begin{aligned} E^{out} &= \{(u, (u, v; t) \mid (u, v; t) \in E^\tau\}, \\ E^{in} &= \{((u, v; t), v) \mid (u, v; t) \in E^\tau\}. \end{aligned}$$

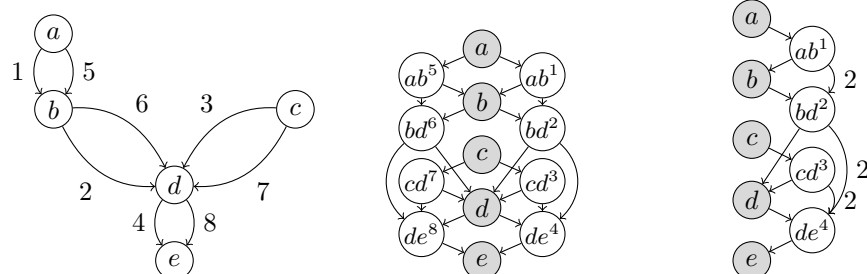

Figure 1: A temporal graph $G^\tau$ (left), the corresponding augmented event graph $G^{\text{aug}}$ (center) and the compressed augmented event graph $G^{\text{comp}}$ (right). In $G^{\text{aug}}$ and $G^{\text{comp}}$, gray nodes have label $0$ and white nodes have label $1$. Timestamped edges $(u, v; t)$ are represented as nodes $uv^t$. In $G^{\text{comp}}$, edge weights represent the number of connected components of $G^{\text{aug}}$ in which the edge appears.

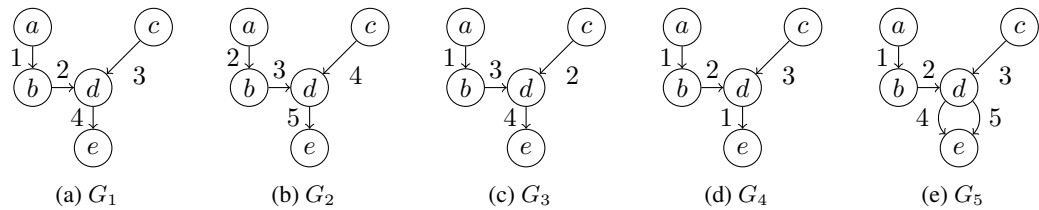

(a) $G_1$      (b) $G_2$      (c) $G_3$      (d) $G_4$      (e) $G_5$

Figure 2: Example illustrating different temporal graph isomorphism definitions for maximum waiting time $\delta = 2$. Edges are labeled with timestamps. $G_1$ is time-concatenated isomorphic to $G_2$, consistent event graph isomorphic to $G_2$ and $G_3$, and time-aggregated isomorphic to $G_2$, $G_3$ and $G_4$.

**Theorem 1.** *(Proof in Appendix A) Let $G_1^\tau$ and $G_2^\tau$ be two temporal graphs with corresponding augmented event graphs $G_1^{aug}$ and $G_2^{aug}$. Then the following statements are equivalent:*

*(i) $G_1^\tau$ and $G_2^\tau$ are time-respecting path isomorphic.*
*(ii) $G_1^\tau$ and $G_2^\tau$ are consistent event graph isomorphic.*
*(iii) $G_1^{aug}$ and $G_2^{aug}$ are isomorphic.*

In addition, we propose to compress the augmented event graph (see Figure 1). This is based on the observation that the temporal event graph often contains many connected components that represent the same set of time-respecting paths, but with different timestamps. Merging these components reduces the size of the graph without losing any information about the causal topology.

**Definition 9** (Compressed augmented event graph). *Let $G^\tau = (V, E^\tau)$ be a temporal graph with its event graph $G^{\mathcal{E}} = (E^\tau, \mathcal{E})$. For each node pair $u, v$, let $t_1, \ldots, t_k$ be the timestamps in $T(u, v)$, sorted in ascending order. The graph $\tau(G^{\mathcal{E}}) = (\tau(E^\tau), \tau(\mathcal{E}))$ replaces each temporal edge $e = (u, v; t_i) \in E^\tau$ with the edge $\tau(e) = (u, v; i)$, i.e., $\tau(E^\tau) = \{\tau(e) \mid e \in E^\tau\}$ and $\tau(\mathcal{E}) = \{(\tau(e), \tau(e')) \mid (e, e') \in \mathcal{E}\}$. Let $\mathcal{C}$ denote the set of connected components in $G^{\mathcal{E}}$. Two connected components $C_1$ and $C_2$ are* equivalent *if $\tau(C_1) = \tau(C_2)$. We compress $G^{\mathcal{E}}$ by replacing each equivalence class of $\mathcal{C}$ with a single representative, in which each edge is weighted with the size of the class. The* compressed augmented event graph *is then built according to Definition 8.*

For reasons of computational efficiency, our notion of equivalence considers the relative order of timestamped edges between the same node pair. As discussed in Appendix B, there are cases in which connected components are not considered equivalent even if their causal topology is the same. In these cases, the compressed augmented event graph does not preserve all isomorphisms.

COMPARISON WITH OTHER ISOMORPHISM DEFINITIONS   We reduced our notion of time-respecting path isomorphism to graph isomorphism on a special static representation of the temporal graph, namely the augmented event graph. We now compare this to isomorphism notions that use different static representations, namely the time-aggregated and time-concatenated static graphs.

**Definition 10** (Time-aggregated/time-concatenated isomorphism). *Let $G_1^\tau$ and $G_2^\tau$ be two temporal graphs with the corresponding time-aggregated graphs $G_1^a$ and $G_2^a$ and the time-concatenated*

*graphs $G_1^c$ and $G_2^c$. We say that $G_1^\tau$ and $G_2^\tau$ are* time-aggregated isomorphic *if $G_1^a$ and $G_2^a$ are isomorphic, and that they are* time-concatenated isomorphic *if $G_1^c$ and $G_2^c$ are isomorphic.*

An equivalent representation of the time-concatenated static graph is a labeled multi-graph in which the timestamps are treated as edge labels. Hence, time-concatenated isomorphism is equivalent to the notions of isomorphism considered in Gao & Ribeiro (2022); Souza et al. (2022). Furthermore, it is similar to the notion of *timewise isomorphism* introduced in Walega & Rawson (2025). The latter was defined for temporal graphs with node labels that may change over time, which are not included in our model. In Appendix C, we show that the two notions are equivalent for temporal graphs without node labels. The following theorem shows that our notion of consistent event graph isomorphism is stricter than time-aggregated isomorphism, but less strict than time-concatenated isomorphism (the proof is given in Appendix C).

**Theorem 2.** *Let $G_1^\tau = (V_1, E_1^\tau)$ and $G_2^\tau = (V_2, E_2^\tau)$ be two temporal graphs with time-aggregated graphs $G_1^a$ and $G_2^a$ and time-concatenated graphs $G_1^c$ and $G_2^c$. Then the following holds:*

*(i) If there exists an isomorphism $\pi_V \colon V_1 \to V_2$ between $G_1^c$ and $G_2^c$, then $\pi_E \colon E_1^\tau \to E_2^\tau$ with*

$$\pi_E(u, v; t) = (\pi(u), \pi(v); t_{min}(G_2^\tau) - t_{min}(G_1^\tau) + t) \quad \forall (u, v; t) \in E_1^\tau$$

*is a consistent event graph isomorphism.*

*(ii) If there exists a consistent event graph isomorphism $\pi_E \colon E_1^\tau \to E_2^\tau$, then the induced node mapping $\pi_V \colon V_1 \to V_2$ is a graph isomorphism between $G_1^a$ and $G_2^a$.*

Time-concatenated isomorphism preserves the values of all timestamps (aside from a constant off-set). By contrast, time-aggregated isomorphism ignores the timestamps altogether. Instead, it only considers the static topology and the number of temporal edges between each node pair. Consistent event graph isomorphism lies between the two: Consider two edges $e = (u, v; t)$ and $e' = (u', v'; t')$. If there is a time-respecting path that includes $e$ before $e'$, then $t < t'$ is enforced. However, if there is no time-respecting path that includes both edges, then the relative order of $t$ and $t'$ is irrelevant.

The differences are illustrated in Figure 2. The graphs $G_1$ and $G_3$ are not time-concatenated isomorphic because the timestamps of the edges $(b, d)$ and $(c, d)$ are flipped, but they are consistent event graph isomorphic because neither graph has a time-respecting path that includes both edges. By contrast, the graphs $G_1$ and $G_4$ are not consistent event graph isomorphic because the path formed by $(b, d; 2)$ and $(d, e; 4)$ is time-respecting in $G_1$ but not in $G_4$. However, they are time-aggregated isomorphic because the static topology and the number of timestamped edges are the same.

## 5 Message Passing for the Augmented Event Graph

We use the equivalency of time-respecting path isomorphism to static isomorphism on the augmented event graph to derive a message-passing GNN architecture for temporal graphs and characterize its expressive power. Note that the augmented event graph is directed, even if the underlying temporal graph is undirected. Edge directions are crucial because they represent the arrow of time, which is why we use the directed GNN Dir-GNN Rossi et al. (2023). It iteratively computes embeddings $f^{(t)}(v)$ for each node $v$ at layer $k$. This is done by aggregating embeddings of its neighbors at layer $k-1$, using $\overrightarrow{f}_{agg}^{(t)}$ for incoming neighbors and $\overleftarrow{f}_{agg}^{(t)}$ for outgoing neighbors. A function $f_{com}^{(t)}$ combines these with the previous embedding of $v$ to a new embedding. Formally, we have

$$f^{(0)}(v) = enc(\ell_V(v)),$$

$$f^{(t)}(v) = f_{com}^{(t)} \left( f^{(t-1)}(v), \overrightarrow{f}_{agg}^{(t)}(\{\{f^{(t-1)}(u) \mid u \in N_I(v)\}\}), \overleftarrow{f}_{agg}^{(t)}(\{\{f^{(t-1)}(u) \mid u \in N_O(v)\}\}) \right)$$

where $f_{com}^{(t)}$, $\overrightarrow{f}_{agg}^{(t)}$ and $\overleftarrow{f}_{agg}^{(t)}$ are learnable. The initial node embeddings are obtained by applying an injective encoding function $enc$ to the node labels. To obtain a representation of the entire graph, we combine embeddings $f^{(k)}(v)$ of all nodes $v$ on the final layer $k$ with an injective readout function.

Our proposed GNN architecture simply applies Dir-GNNs to the augmented event graph. By using the augmented event graph, this approach is specifically tailored towards detecting time-respecting path isomorphism. We give an informal analysis of its expressive power (further details in Appendix D): A model $M_1$ is *at least as expressive* as another model $M_2$ if $M_1$ distinguishes all node

pairs that are distinguished by $M_2$. If the reverse is also true, they are *equally as expressive*. Otherwise, $M_1$ is *strictly more expressive*. Rossi et al. (2023) prove that if $f_{\text{com}}^{(t)}$, $\overrightarrow{f}_{\text{agg}}^{(t)}$ and $\overleftarrow{f}_{\text{agg}}^{(t)}$ are injective, Dir-GNN has the same expressive power as a directed version of the 1-WL test, called D-WL. Furthermore, both are strictly more expressive than undirected 1-WL and GNNs, i.e., there are graphs in which Dir-GNNs and D-WL distinguish more nodes than 1-WL and GNNs.

Finally, we note the similarity of our approach to the temporal WL graph kernel proposed in Oettershagen et al. (2020). Their approach applies 1-WL to the *directed line graph expansion*, which is equivalent to applying D-WL to the (unaugmented) event graph. This does not take into account the node consistency property (Definition 7). Furthermore, we proposed a GNN architecture, which offers more flexibility since it can learn the combination and aggregation functions.

## 6 EXPERIMENTAL EVALUATION

We evaluate the message passing architecture proposed in the previous section in a graph classification experiment. With this we seek to answer the following research questions:

**(RQ1)** Does the message passing scheme for temporal event graphs derived in Section 4 allow to distinguish non-isomorphic temporal graphs in a graph classification experiment.

**(RQ2)** How does our model compare against existing methods in empirical datasets

To address **RQ1**, we evaluate the accuracy of our model for classifying synthetically generated temporal graphs that *exclusively differ* in terms of their *causal topology*, i.e. which nodes are connected via time-respecting paths. We use two stochastic models generating temporal graphs that are indistinguishable based on their time-aggregated static graphs (cf. Definition 5).

**Experiment A: Shuffled Timestamps** We generate temporal graphs from random walks on a $k$-regular graph and in a second step randomly permute timestamps for a fraction $\alpha$ of edges. The original graphs and shuffled versions then share the same time-aggregated topology but differ in causal structure. These two variants define the classes for our binary classification task. Details of graph size, walk generation, and timestamp assignment are given in Appendix E

**Exeeriment B: Cluster Connectivity** Following Scholtes et al. (2014), we build temporal graphs with two communities linked by few inter-edges. By varying a parameter $\sigma$, we control whether cross-cluster time-respecting paths are over- ($\sigma > 0$) or underrepresented ($\sigma < 0$), while time-aggregated structure remains identical. Graphs generated with $\sigma = 0$ and those generated with a $\sigma \neq 0$ form the two classes. The exact construction procedure is described in Appendix E.

For all experiments, we address a balanced, binary temporal graph classification task, i.e. given $n$ temporal graphs $G_i^\tau$ ($i = 1, \dots, n$) we want to learn a classifier $C \colon \{G_i^\tau\} \to \{0, 1\}$, with ground truth classes assigned as described above. Importantly, for a $G_i^\tau$ we predict a *single class* rather than multiple classes for different times $t$ in the evolution of $G_i^\tau$. Details are in Appendix E

**Discussion** Figure 3 (left panel) shows mean accuracies of our TGNN (y-axis) in experiment A for different fractions $\alpha$ of shuffled timestamps (x-axis) across 100 runs. Our model classifies temporal graphs with near-perfect accuracy for $\alpha > 0.2$, while accuracy decreases for $\alpha \approx 0$. The mean accuracy (100 runs) for experiment B (middle panel) show that our model is able to distinguish the patterns in time-respecting paths generated for different values of $\sigma$ (x-axis). Accuracy (y-axis) increases as $\sigma$ deviates from the baseline $\sigma = 0$, for which graphs are –by definition– indistinguishable. The right panel of Figure 3 shows mean accuracies of our TGNN where the two classes are defined by different $\sigma_1$ and $\sigma_2$ (standard deviation in Figure 5 in Appendix F). Our model reliably distinguishes graphs with random patterns from non-random patterns, independent of whether cross-cluster time-respecting paths are over- ($\sigma > 0$) or under-represented ($\sigma < 0$).

To address **RQ2**, we compare our method against two baselines: a Graph Attention Network (GAT) Veličković et al. (2018) applied to time-concatenated static graphs with timestamps as edge features, and the Temporal Graph Network (TGN) Rossi et al. (2020), following the temporal graph classification setup of Gao & Ribeiro (2022). We evaluate on five real-world datasets where class differences arise solely from time-respecting paths rather than static topology. To create binary classification tasks, we shuffle edge timestamps for one class and generate additional graphs by applying small timestamp perturbations. Table 1 shows that our method substantially outperforms the baselines and

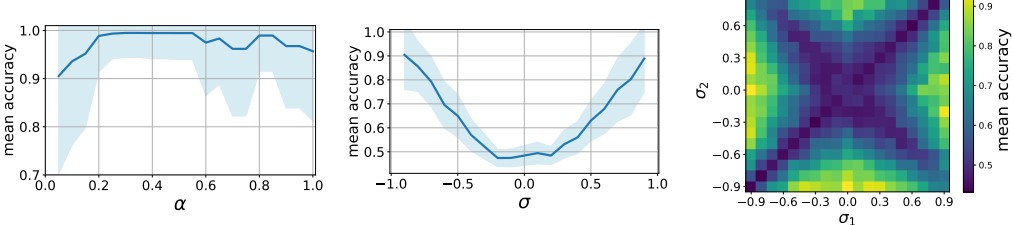

Figure 3: Results of classification experiments A (left) and B (middle) where we use our TGNN model to classify temporal graphs with different $\alpha$ (left) and $\sigma$ (right) (x-axis), respectively. Results are averaged over 100 runs. Right panel: mean classification accuracy for temporal graphs generated with $\sigma_1$ vs. $\sigma_2$ (for all pairs $\sigma_1, \sigma_2$, 25 runs each).

achieves near perfect accuracies. We additionally implemented the shuffling approach from Pritam et al. (2025) , in particular adopting the randomized edge (RE) and the configuration model (CM) classes for our real-world data sets. The accuracy results are given in Table 4

We did not include PINT or CAW as baselines because they are designed for link prediction. Adapting them for graph classification would require pooling node embeddings at the last timestamp, but their recency-weighted aggregation biases the graph embedding toward the most recent events, whereas classification requires equally weighted information from the entire history.

| Data | | Our Model | GAT | TGN |
|---|---|---|---|---|
| ants-1-1 | Blonder & Dornhaus (2011) | **0.87 ± 0.18** | 0.52 ± 0.04 | 0.86 ± 0.05 |
| ants-1-2 | Blonder & Dornhaus (2011) | 0.93 ± 0.09 | 0.49 ± 0.08 | **0.99 ± 0.02** |
| sp-workplace | Génois & Barrat (2018) | **0.98 ± 0.02** | 0.48 ± 0.07 | 0.51 ± 0.05 |
| sp-hospital | Vanhems et al. (2013) | **1.00 ± 0.00** | 0.48 ± 0.08 | 0.82 ± 0.06 |
| eu-email-dept2 | Paranjape et al. (2017) | **0.85 ± 0.20** | 0.58 ± 0.07 | 0.56 ± 0.06 |

Table 1: Mean classification accuracy on real-world data. For each dataset, we split the timeline into windows of 500 timestamps, performed 10 runs per window, and report the average over all runs.

**Ablation study** We perform an ablation study to test which components of our proposed architecture are necessary. The results are reported in Appendix G.

## 7 CONCLUSION

We theoretically investigate the expressivity of temporal graph neural networks (TGNN). We introduce a natural generalization of graph isormorphism to temporal graphs by considering how the arrow of time shapes time-respecting paths and thus the causal topology of temporal graphs. We show that this isomorphism can be heuristically tested by applying the directed and labeled Weisfeiler-Leman algorithm to augmented temporal event graphs. This suggests a neural message passing architecture that is expressive enough to distinguish temporal graphs with identical static topology but different time-respecting paths. We evaluate our model in synthetic temporal graphs.

**Limitations and Open Issues** A limitation of our work is that we did not perform a comprehensive comparative evaluation of our approach to other TGNN architectures. However these often do not naturally lend themselves to the temporal graph classification task addressed in this work. E.g. memory-based models like TGN Rossi et al. (2020) are commonly trained using batches of timestamped edges from a single temporal graph, making it non-trivial to adapt them for training on batches containing multiple temporal graphs as done in our work. This adaptation of existing TGNNs goes beyond the scope of our work, which is why we leave it for future work. Focusing on a theoretical investigation of expressivity, we further did not perform experiments in other real-world graph learning tasks, where ground truth is available, such as node classification or link prediction. We also did not formally compare our definition of temporal graph isomorphism to the notion of isomorphism on temporal computation trees, which is the basis for the analysis by Souza et al. (2022). This is an interesting open question and an example for the research avenues opened by our work.

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

## A  EQUIVALENCE OF TEMPORAL GRAPH ISOMORPHISM NOTIONS

In the following we give the proof of Theorem 1.

**Theorem.** *Let $G_1^\tau = (V_1, E_1^\tau)$ and $G_2^\tau = (V_2, G_2^\tau)$ be two temporal graphs with corresponding augmented event graphs $G_1^{aug} = (V_1^{aug}, E_1^{aug}, \ell_1)$ and $G_2^{aug} = (V_2^{aug}, E_2^{aug}, \ell_2)$. Then the following statements are equivalent:*

*(i)  $G_1^\tau$ and $G_2^\tau$ are time-respecting path isomorphic.*
*(ii)  $G_1^\tau$ and $G_2^\tau$ are consistent event graph isomorphic.*
*(iii)  $G_1^{aug}$ and $G_2^{aug}$ are isomorphic.*

We begin by showing the equivalence of (i) and (ii):

*Proof.* Let $\pi_V \colon V_1 \to V_2$ and $\pi_E \colon E_1^\tau \to E_2^\tau$ be a node and edge mapping, respectively. Let $p = (v_0, e_1, v_1, \ldots, e_{k-1}, v_k)$ be an alternating sequence of nodes and timestamped edges in $G_1^\tau$. We denote the corresponding sequence in $G_2^\tau$ that is induced by $\pi_V$ and $\pi_E$ as $\pi(p) = (\pi_V(v_0), \pi_E(e_1), \pi_V(v_1), \ldots, \pi_E(e_{k-1}), \pi_V(v_k))$. We say that $\pi_V$ and $\pi_E$ are *path-preserving* between $G_1^\tau$ and $G_2^\tau$ if for each sequence $p$ as defined above, $p$ is a path in $G_1^\tau$ if and only if $\pi(p)$ is a path in $G_2^\tau$. It is easy to see that $\pi_V$ and $\pi_E$ are path-preserving between $G_1^\tau$ and $G_2^\tau$ if and only if $\pi_E$ is node-consistent with $\pi_V$.

Assume therefore that $\pi_V$ and $\pi_E$ are path-preserving between $G_1^\tau$ and $G_2^\tau$. We show that $\pi_E$ is a graph isomorphism between the temporal event graphs $G_1^\mathcal{E}$ and $G_2^\mathcal{E}$ if and only if it is *time-preserving*, i.e., a path $p$ in $G_1^\tau$ is time-respecting iff $\pi(p)$ is time-respecting in $G_2^\tau$. If $k = 1$, this holds trivially because all paths of length 1 are time-respecting. If $k \geq 2$, then $p$ is time-respecting if and only if $(e_1, (e_1, e_2), e_2, \ldots, (e_{k-2}, e_{k-1}), e_{k-1})$ is a path in $G_1^\mathcal{E}$. Hence, $\pi_E$ is time-preserving if and only if it is path-preserving between $G_1^\mathcal{E}$ and $G_2^\mathcal{E}$. Because the event graphs are unlabeled, this is the case if and only if $\pi_E$ is a graph isomorphism by Observation 1. $\square$

Next, we show the equivalence of (ii) and (iii):

*Proof.* Let $\pi \colon V_1^{\mathrm{aug}} \to V_2^{\mathrm{aug}}$ be an isomorphism between $G_1^{\mathrm{aug}}$ and $G_2^{\mathrm{aug}}$. Because $\pi$ preserves the node labels, it can be decomposed into bijective mappings $\pi_V \colon V_1 \to V_2$ and $\pi_E \colon E_1^\tau \to E_2^\tau$. Then $\pi_E$ is an isomorphism between $G_1^\mathcal{E}$ and $G_2^\mathcal{E}$ because these are subgraphs of $G_1^{\mathrm{aug}}$ and $G_2^{\mathrm{aug}}$, respectively. Consider an edge $e = (u, v; t) \in E_1^\tau$. By construction, $G_1^{\mathrm{aug}}$ includes the edges $(u, e) \in E_1^{\mathrm{out}}$ and $(e, v) \in E_1^{\mathrm{in}}$. Because $\pi$ is an isomorphism, it follows that $(\pi_V(u), \pi_E(e)) \in E_2^{\mathrm{out}}$ and $(\pi_E(e), \pi_V(v)) \in E_2^{\mathrm{in}}$. Then it follows by construction of $G_2^{\mathrm{aug}}$ that $\pi_E(e) = (\pi_V(u), \pi_V(v); t')$ for some $t' \in \mathbb{N}$.

Conversely, let $\pi_E \colon E_1^\tau \to E_2^\tau$ be a consistent event graph isomorphism between $G_1^\tau$ and $G_2^\tau$, and let $\pi_V \colon V_1 \to V_2$ be the induced node mapping such that

$$\forall (u, v; t) \in E_1^\tau \quad \exists t' \colon \pi_E(u, v; t) = (\pi_V(u), \pi_V(v); t').$$

Then $\pi_E$ and $\pi_V$ can be combined into a bijective mapping $\pi \colon V_1^{\mathrm{aug}} \to V_2^{\mathrm{aug}}$. We show that $\pi$ is an isomorphism between $G_1^{\mathrm{aug}}$ and $G_2^{\mathrm{aug}}$. By construction, $\pi$ preserves the node labels. For every pair of nodes $x, y \in V_1^{\mathrm{aug}}$ and every set of edges $E' \in \{\mathcal{E}, E^{\mathrm{out}}, E^{\mathrm{in}}\}$, we show that

$$(x, y) \in E_1' \quad \Longleftrightarrow \quad (\pi(x), \pi(y)) \in E_2'.$$

For $E' = \mathcal{E}$, this follows from the fact that $\pi_E$ is an isomorphism between $G_1^\mathcal{E}$ and $G_2^\mathcal{E}$. We show the case $E' = E^{\mathrm{out}}$ (the case $E' = E^{\mathrm{in}}$ is symmetrical): We have $(u, y) \in E_1^{\mathrm{out}}$ if and only if $y = (x, v; t)$ for some $v \in V_1$ and $t \in \mathbb{N}$. We have $\pi(y) = \pi_E(y) = (\pi_V(x), \pi_V(v); t') = (\pi(x), \pi(v); t')$ for some $t' \in \mathbb{N}$. By definition of $E^{\mathrm{out}}$, we have $(\pi(x), \pi(y)) \in E_2^{\mathrm{out}}$. $\square$

## B  TEMPORAL EVENT GRAPH COMPRESSION

Figure 4 shows an example in which a consistent event graph isomorphism between two temporal graphs $G_1^\tau$ and $G_2^\tau$ is lost when compressing the temporal event graph. In the temporal event graph of both $G_1^\tau$ and $G_2^\tau$, all four connected components are isomorphic, but there are two equivalence

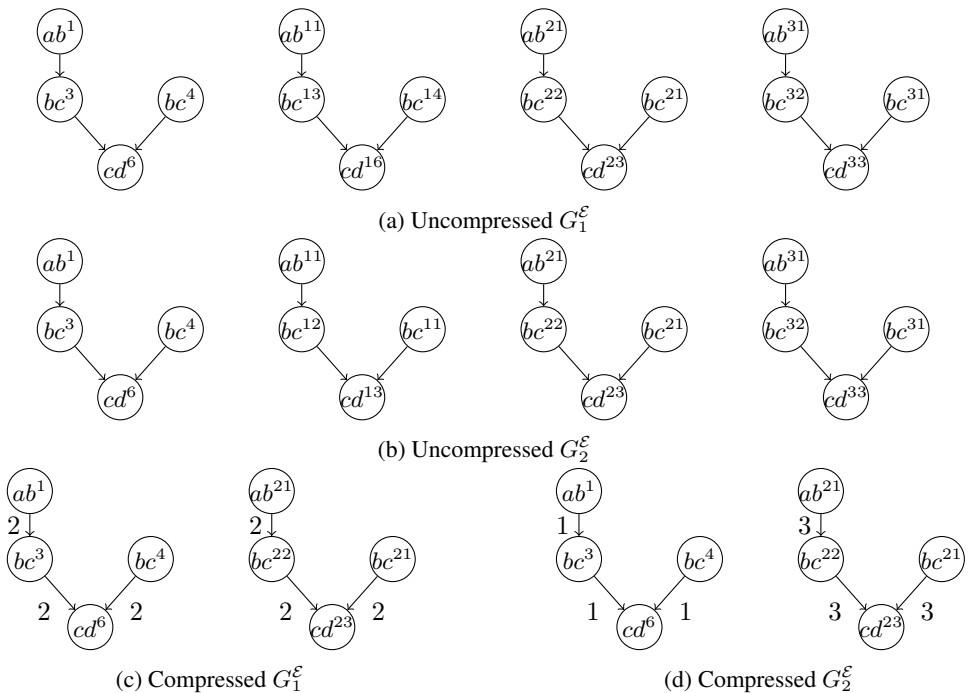

(a) Uncompressed $G_1^{\mathcal{E}}$

(b) Uncompressed $G_2^{\mathcal{E}}$

(c) Compressed $G_1^{\mathcal{E}}$        (d) Compressed $G_2^{\mathcal{E}}$

Figure 4: An example of two temporal graphs $G_1^\tau$ and $G_2^\tau$ which are consistent event graph isomorphic, but there is no node-consistent isomorphism between the compressed event graphs.

classes. This is due to the relative order of the two $(b, c)$ edges: in some components, the edge that is adjacent to $(a, b)$ has an earlier timestamp than the other one, but in other components the order is flipped. The cardinality of the equivalence classes differs between $G_1^{\mathcal{E}}$ and $G_2^{\mathcal{E}}$, and therefore the edge weights of the representatives in the compressed event graphs differs. Therefore, there is no node-consistent isomorphism between the compressed event graphs, even though there is one between the uncompressed ones.

This phenomenon is due to the fact that the definition of equivalence between connected components considers the relative order of timestamped edges that connect the same node pair. This is done to ensure that connected components can be tested for equivalence efficiently. As Figure 4 shows, it is possible for two connected components to be isomorphic even if the relative order is different. Hence, the notion of consistent event graph isomorphism becomes slightly stricter if the event graphs are compressed. As a result, a WL test or GNN architecture using event graph compression may distinguish some graphs that should not be distinguished. We nevertheless accept this loss in precision because the compression can significantly reduce the size of the graph and thereby makes the GNN easier to train.

## C  COMPARISON WITH OTHER GRAPH ISOMORPHISMS

In the following we give the proof of Theorem 2.

**Theorem.** *Let $G_1^\tau = (V_1, E_1^\tau)$ and $G_2^\tau = (V_2, E_2^\tau)$ be two temporal graphs with time-aggregated graphs $G_1^a = (V_1, E_1^s, \ell_1^a)$ and $G_2^a = (V_2, E_2^s, \ell_2^a)$ and time-concatenated graphs $G_1^c = (V_1, E_1^s, \ell_1^c)$ and $G_2^c = (V_2, E_2^s, \ell_2^c)$. Then the following holds:*

*(i) If there exists an isomorphism $\pi_V \colon V_1 \to V_2$ between $G_1^c$ and $G_2^c$, then $\pi_E \colon E_1^\tau \to E_2^\tau$ with*

$$\pi_E(u, v; t) = (\pi(u), \pi(v); t_{min}(G_2^\tau) - t_{min}(G_1^\tau) + t) \quad \forall (u, v; t) \in E_1^\tau$$

*is a consistent event graph isomorphism.*

*(ii) If there exists a consistent event graph isomorphism $\pi_E \colon E_1^\tau \to E_2^\tau$, then the induced node mapping $\pi_V \colon V_1 \to V_2$ is a graph isomorphism between $G_1^a$ and $G_2^a$.*

*Proof.* Claim (i) follows directly from the definitions. For claim (ii), let $\pi_E \colon E_1^\tau \to E_2^\tau$ be a consistent event graph isomorphism. For every node pair $u, v \in V_1$, we have

$$
\begin{aligned}
|T_1(u, v)| &= |\{(u, v; t) \in E_1^\tau\}| \\
&= |\{\pi_E(u, v; t) \in E_2^\tau\}| \\
&= |\{(\pi_V(u), \pi_V(v); t) \in E_2^\tau\}| \\
&= |T_2(\pi_V(u), \pi_V(v))|.
\end{aligned}
$$

Here, we use the node consistency property and the fact that $\pi_E$ is a bijection. It follows that

$$
\begin{aligned}
(u, v) \in E_1^{\mathrm{s}} &\iff T_1(u, v) \neq \emptyset \\
&\iff T_2(\pi_V(u), \pi_V(v)) \neq \emptyset \\
&\iff (\pi_V(u), \pi_V(v)) \in E_2^{\mathrm{s}}
\end{aligned}
$$

and

$$
\ell_1^{\mathrm{a}}(u, v) = |T_1(u, v)| = |T_2(\pi_V(u), \pi_V(v))| = \ell_2^{\mathrm{a}}(\pi_V(u), \pi_V(v)). \qquad \square
$$

### C.1 Timewise Isomorphism with Time-Variant Node Labels

Walega and Rawson Walega & Rawson (2025) present isomorphism definitions for snapshot-based temporal graphs in which the nodes have labels that can vary over time.

**Definition 11** (Snapshot-based temporal graph). *A snapshot-based temporal graph is a sequence $(G_1, t_1), \ldots, (G_n, t_n)$ with $t_1 < \cdots < t_n$ and $G_i = (V, E_i, c_i)$, where $c_i \colon V \to \mathcal{L}_V$ is a node labeling.*

**Definition 12** (Timewise isomorphism). *Two snapshot-based temporal graphs $(G_1, t_1), \ldots, (G_n, t_n)$ and $(\hat{G}_1, \hat{t}_1), \ldots, (\hat{G}_{\hat{n}}, \hat{t}_{\hat{n}})$ are timewise isomorphic if $n = \hat{n}$, $t_i - t_1 = \hat{t}_i - \hat{t}_1$ for all $1 \leq i \leq n$, and there is a bijection $\pi \colon V \to V'$ such that $\pi$ is an isomorphism between $G_i$ and $\hat{G}_i$ for all $1 \leq i \leq n$.*

We consider the case that there are no node labels. In this case, a snapshot-based temporal graph $(G_1, t_1), \ldots, (G_n, t_n)$ has an equivalent temporal graph $G^\tau = (V, E^\tau)$ according to our definition with

$$
E^\tau = \{(u, v; t_i) \mid (u, v) \in E_i, 1 \leq i \leq n\}.
$$

The following theorem shows that if there are no node labels, timewise isomorphism is equivalent to isomorphism of the time-concatenated static graphs.

**Theorem 3.** *Let $(G_1, t_1), \ldots, (G_n, t_n)$ and $(\hat{G}_1, t_1), \ldots, (\hat{G}_n, t_n)$ be two snapshot-based temporal graphs without node labels, and let $\pi \colon V \to \hat{V}$ be a node bijection. Then $\pi$ is an isomorphism between the time-concatenated static graphs $G^c = (V, E^{\mathrm{s}}, \ell^c)$ and $\hat{G}^c = (\hat{V}, \hat{E}^{\mathrm{s}}, \hat{\ell}^c)$ iff $n = \hat{n}$, $t_i - t_1 = \hat{t}_i - \hat{t}_1$ for all $1 \leq i \leq n$, and $\pi$ is an isomorphism between $G_i$ and $\hat{G}_i$ for all $1 \leq i \leq n$.*

*Proof.* For each node pair $u, v \in V$, we have

$$
T(u, v) = \{t - t_{\min}(G^\tau) \mid (u, v; t) \in E^\tau\} = \{t_i - t_1 \mid (u, v) \in E_i, 1 \leq i \leq n\}.
$$

We note that $\pi$ is an isomorphism between $G^c$ and $\hat{G}^c$ iff $T(u, v) = \hat{T}(\pi(u), \pi(v))$ holds for all $u, v \in V$. If $n = \hat{n}$, $t_i - t_1 = \hat{t}_i - \hat{t}_1$ for all $1 \leq i \leq n$, and $\pi$ is an isomorphism between $G_i$ and $\hat{G}_i$ for all $1 \leq i \leq n$, then it is easy to see that $T(u, v) = \hat{T}(\pi(u), \pi(v))$ for all $u, v \in V$. On the other hand, assume that $T(u, v) = \hat{T}(\pi(u), \pi(v))$ for all $u, v \in V$. We have

$$
\begin{aligned}
(u, v) \in E_i &\iff t_i - t_1 \in T(u, v) \iff t_i - t_1 \in \hat{T}(\pi(u), \pi(v)) \\
&\iff \exists j \colon \hat{t}_j - \hat{t}_1 = t_i - t_1 \wedge \hat{t}_j - \hat{t}_1 \in \hat{T}(\pi(u), \pi(v)) \\
&\iff \exists j \colon \hat{t}_j - \hat{t}_1 = t_i - t_1 \wedge (\pi(u), \pi(v)) \in \hat{E}_j.
\end{aligned}
$$

Within each temporal graph, all timestamps are distinct from each other, so by the pigeonholing principle, we have $n = \hat{n}$ and $t_i - t_1 = \hat{t}_j - \hat{t}_1$ for all $1 \leq i \leq n$. Then it follows that $\pi$ is an isomorphism between $G_i$ and $\hat{G}_i$ for all $1 \leq i \leq n$. $\qquad \square$

# D EXPRESSIVE POWER OF MESSAGE PASSING ON THE AUGMENTED EVENT GRAPH

We formally analyze the expressive power of the message passing approach presented in Section 5, which applies Dir-GNN to the augmented event graph. Rossi et al. Rossi et al. (2023) propose a directed version of the 1-WL test, called D-WL. It is a *color refinement algorithm* that iteratively computes a node coloring $c^{(t)} \colon V \to \mathbb{N}$ for each iteration $t \geq 0$ as follows:

$$c^{(0)}(v) = \ell_V(v)$$

$$c^{(t)}(v) = \text{hash}\left(c^{(t-1)}(v), \{\!\{c^{(t-1)}(u) \mid u \in N_I(v)\}\!\}, \{\!\{c^{(t-1)}(u) \mid u \in N_O(v)\}\!\}\right).$$

Here, hash is an injective function. Compared to the standard 1-WL test, D-WL takes edge directions into account by considering the multisets of node colors of the incoming and outgoing neighbors separately.

The node embeddings computed by GNNs can also be interpreted as node colorings, and thus GNNs can be seen as color refinement algorithms. The ability of color refinement algorithms to distinguish nodes is captured by the following definition.

**Definition 13** (Expressivity). *A color refinement algorithm $c$ is at least as expressive as another color refinement algorithm $\hat{c}$ if for all graphs $G$, all iterations $t \geq 0$ and all node pairs $u, v$, it holds that $\hat{c}^{(t)}(u) \neq \hat{c}^{(t)}(v)$ implies $c^{(t)}(u) \neq c^{(t)}(v)$. If the reverse implication holds as well, $c$ and $\hat{c}$ are equally as expressive. Otherwise, $c$ is strictly more expressive than $\hat{c}$.*

Morris et al. Morris et al. (2019) show that undirected GNNs are equally as expressive as undirected 1-WL if the aggregation and combination functions are injective. Rossi et al. Rossi et al. (2023) extend this result to the directed variants: Dir-GNN is equally as expressive as D-WL if $f_{\text{com}}^{(t)}$, $\overrightarrow{f}_{\text{agg}}^{(t)}$ and $\overleftarrow{f}_{\text{agg}}^{(t)}$ are injective. Furthermore, both are strictly more expressive than undirected 1-WL and GNNs, i.e., there are graphs in which Dir-GNNs and D-WL distinguish more nodes than 1-WL and GNNs. Because the augmented event graph is a static directed graph, these results automatically carry over to our setting:

**Theorem 4.** *If $f_{\text{com}}^{(t)}$, $\overrightarrow{f}_{\text{agg}}^{(t)}$ and $\overleftarrow{f}_{\text{agg}}^{(t)}$ are injective, Dir-GNN on the augmented event graph has the same expressive power as D-WL on the augmented event graph.*

# E EXPERIMENTAL PROTOCOL

In the following, we give a detailed overview of our experimental protocol.

**Experiment A: Shuffled Timestamps** We first use a simple shuffling model to generate temporal graphs: Starting from a static $k$-regular random graph ($n = 10$, $k = 3$), we generate 250 temporal graphs by simulating 500 random walks with length two for each of them. Traversed edges are assigned consecutive timestamps $t$ and $t + 1$ within each walk, with an additional time gap of $t + 2$ separating different walks. This leads to 250 temporal graphs with 1000 time-stamped edges, each containing 500 time-respecting paths of lengths two (for $\delta = 1$). For each graph we then randomly permute timestamps for a fraction $\alpha$ of edges. This does not change edge frequencies, i.e. it yields the same time-aggregated weighted graph. However, due to the arrow of time, shuffling the timestamps of edges affects time-respecting paths and thus the causal topology. We assign original temporal graphs to one class and graphs with shuffled timestamps to another class.

**Experiment B: Cluster Connectivity** In a second experiment we adopt the stochastic model proposed in Scholtes et al. (2014). This model is based on a static graph with two strong communities, each consisting of a $k$-regular random graph with $k = 3$ and $n_1 = n_2 = 10$ nodes that are interconnected by two edges. Similar to experiment A, we randomly generate temporal graphs by simulating 500 second-order random walks of length two (using the same approach to assign timestamps). A parameter $\sigma \in (-1, 1)$ allows to tune random walk transition probabilities such that (i) for $\sigma < 0$ time-respecting paths between *different* communities are underrepresented compared to a shuffled temporal graph, (ii) for $\sigma > 0$ time-respecting paths connecting different communities are *overrepresented*, and (iii) for all $\sigma$ time-aggregated weighted graphs are identical (see (Scholtes et al.,

|  | Our model | GAT | TGN |
|---|---|---|---|
| batch size | 200 | 50 | 200 |
| weight_decay | 0.0001 | 0 | 0 |
| learning rate | 0.001 | 0.001 | 0.0001 |
| DirGNN layers | $32 \to 16 \to 8$ | – | – |
| GatConv Layers | – | $8 \to 64 \to 64$ | – |
| dense layers | $8 \to 4 \to 1$ | $64 \to 1$ | $100 \to 1$ |
| Pooling | global add | global mean | global add |
| node feature dim | 2 (OHE of node labels) | n (OHE of nodes) | |
| tunable parameters | 3777 | 5825 | 192101 |

Table 2: Optimal hyperparameter values and number of tunable model parameters (i.e. model size) for the experiments using temporal graphs generated by Model A and Model B.

| Library/Software | Version |
|---|---|
| CUDA | cu121 |
| `torch` | 2.4.1+cu121 |
| `torch_cluster` | 1.6.3+pt24cu121 |
| `torch_scatter` | 2.1.2+pt24cu121 |
| `torch_sparse` | 0.6.18+pt24cu121 |
| `torch_geometric` | 2.5.1 |
| `pyg-lib` | 0.4.0+pt24cu121 |
| `pathpyG` | 0.2.0 |

Table 3: Version of key dependencies used in the implementation of our experimental evaluation

2014)). This model generates random temporal graphs that share the same time-aggregated static topology, but whose causal topology differs in terms of how well nodes in different communities are connected via time-respecting paths. We use this model to generate 250 temporal graphs for different $\sigma$, assigning graphs generated with $\sigma = 0$ to one class and those generated with $\sigma \neq 0$ to another class.

We used the temporal graph learning library `pathpyG` to implement the random walk based models for synthetic temporal graphs (Experiment A and B) explained in Section 6. We then used `pathpyG` to generate compressed augmented event graphs for $\delta = 1$ for all temporal graphs generated by the two models.

We implemented the TGNN from Section 5 using `pytorch-geometric` (pyG) Fey & Lenssen (2019). For message passing in the compressed augmented event graph, we use three convolutional layers (with layer widths being hyperparameters) using pyG's implementation of Dir-GNN (using the `GraphConv` layer) proposed in Rossi et al. (2023). We use an `add` pooling layer and two dense linear classification layers, where layer widths are hyperparameters. For all experiments, we address a balanced, binary temporal graph classification task, i.e. given $n$ temporal graphs $G_i^\tau$ ($i = 1, \ldots, n$) we want to learn a classifier $C \colon \{G_i^\tau\} \to \{0, 1\}$, with ground truth classes assigned as described above. Importantly, for a $G_i^\tau$ we predict a *single class* rather than multiple classes for different times $t$ in the evolution of $G_i^\tau$. We use a single output with sigmoid activation in the final layer and train the model using binary cross entropy loss. We use a one-hot encoding of node labels in the augmented event graph (cf. $\ell_V$ in Definition 8) as node features.

We used the binary cross entropy loss function.

For all experiments we ran the Adam optimizer with a weight decay of $0.0001$ for 250 epochs with a 80/20 training/test split, and a fixed batch size of 200 (temporal graphs). We performed a grid search to tune hyperparameters (learning rate, width of GNN and dense classification layers). The optimal hyperparameters used to obtain the results for experiment A, experiment B as well as the comparison to the GAT model applied to the time-concatenated static graph as described in Section 6 are reported in Table 2.

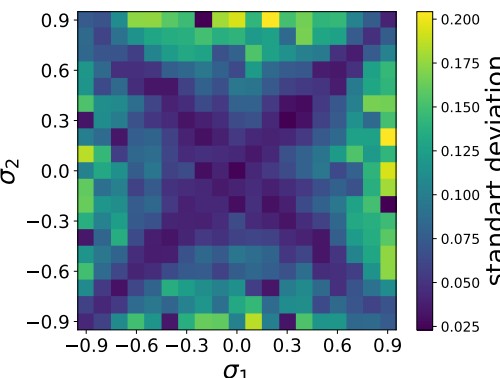

Figure 5: Standard deviation of accuracies of our TGNN model for classification of temporal graphs with $\sigma_1$ vs. $\sigma_2$ for all pairs of $\sigma_1, \sigma_2$ (associated with right panel of Figure 3).

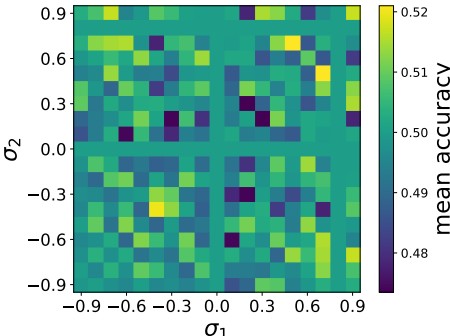

Figure 6: Mean accuracy of accuracies of a GAT model for classification of temporal graphs with $\sigma_1$ vs. $\sigma_2$ for all pairs of $\sigma_1, \sigma_2$. The mean standard deviation across all runs is $0.048$.

More detailed information on library versions, parameter number of our models, as well as the computational resources used during training are included in the appendix. In the appendix, we further report the optimal hyperparameter values found in the grid search. Upon acceptance of our manuscript, we will make the code of our experiments as well as a container description available to ensure the reproducibility of our results.

We ran our experiments on a container-based (Singularity) HPC environment with 4.512 CPU cores and 160 GPUs (122 x L40, 24 x L40s, 16 x H100). For our experiments we used a total of less than 30 GPU hours for training and evaluation.

## F  ADDITIONAL RESULTS

In this section, we include additional results for the temporal graph classification experiment B for parameter pairs $\sigma_1, \sigma_2$, where we assign all temporal graphs generated for $\sigma_1$ to one class, while all temporal graphs generated for $\sigma_2$ are assigned to the other class. In Figure 5 we show the standard deviation of classification accuracies of our TGNN model, fitting the mean accuracies of our model reported in the right panel of Figure 3 in Section 6.

The left panel of Figure 6 shows the mean accuracy of Graph Attention Network applied to the time-concatenated static graph across 20 runs (mean standard deviation $0.048$). We find that this model is not able to reliably classify temporal graphs for any combination of parameters $\sigma_1, \sigma_2$.

Table 4 reports the additional results of our model and TGN on the real world datasets obtained by adopting the shuffling approaches by Pritam et al. (2025)

| Data | Our Model | TGN |
|------|-----------|-----|
| ants-1-1 | $\mathbf{1.00 \pm 0.00}$ | $0.97 \pm 0.02$ |
| ants-1-2 | $\mathbf{1.00 \pm 0.00}$ | $0.92 \pm 0.01$ |
| sp-workplace | $\mathbf{1.00 \pm 0.00}$ | $0.74 \pm 0.11$ |
| sp-hospital | $\mathbf{1.00 \pm 0.00}$ | $0.82 \pm 0.12$ |
| eu-email-dept2 | $\mathbf{1.00 \pm 0.00}$ | $0.96 \pm 0.02$ |

Table 4: Mean classification accuracy on real-world datasets with classes obtaine from randomized edges and configuration model (Pritam et al. (2025) For each dataset, we split the timeline into windows of 500 timestamps, performed 10 runs per window, and report the average over all runs.

| Experiment | Accuracy |
|------------|----------|
| Compressed augmented event graph | $0.95 \pm 0.03$ |
| No node labels | $0.66 \pm 0.20$ |
| Message passing only in one direction | $0.63 \pm 0.18$ |
| No edge weights | $0.81 \pm 0.22$ |

Table 5: Results of ablation study for experiment B (classification of temporal graphs generated for $\sigma = 0$ vs. $\sigma = 0.9$).

# G  ABLATION STUDY

Table 5 shows results of an ablation study, where we selectively remove those aspects of our message passing architecture, which –based on our theoretical analysis in Section 5– we predict to be necessary to distinguish non-isomorphic temporal graphs. We first remove node labels that indicate whether nodes in the augmented event graph represent nodes or time-stamped edges in the original temporal graph (cf. $\ell_V$ in Definition 8). We then remove the bidirectional message passing discussed in Section 5. We finally ignore edge weights in the compressed event graph, as introduced in Section 4. The results in Table 5 demonstrate that the message passing architecture proposed in Section 5, which integrates all of those components, achieves the best performance (top-most row) compared to other approaches (other rows).

