# OpenReview forum: "Weisfeiler and Leman Follow the Arrow of Time: Expressive Power of Message Passing in Temporal Event Graphs"
_ICLR.cc/2026/Conference — ICLR 2026 Conference Withdrawn Submission_

### Official Review · Reviewer_Hcq2 · 2025-10-23

**Soundness:** 2
**Presentation:** 3
**Contribution:** 3
**Rating:** 6
**Confidence:** 4

**Summary:**

In this work, the authors proposed a novel notion of expressiveness for TGNNs in the form of consistent event graph isomorphism, which is based on time-respecting paths in temporal graphs. This expressiveness can be measured by a temporal generalization of the WL algorithm to determine non-isomorphic temporal graphs. A novel message passing architecture is proposed to be expressive enough to distinguish TGs with identical static topology yet differing time-respecting paths. This is evaluated on synthetic tasks as well as temporal graph classification task.

**Strengths:**

Overall, I liked the novel graph isomorphism formulation introduced in this work and focusing on aspects currently under-explored in the literature, i.e. time-respecting paths. The strengths of the paper is listed as follows. However, there are some weaknesses in this work (see next Section) and thus I am currently more neutral on the work, leaning weak accept.

- **a novel graph isomorphism for temporal graphs**. The authors proposed a novel temporal generalisation of graph isomorphism called time-respecting path isomorphism which focuses on the the concept of time-respecting path.

- **clear presentation**. The ideas in the paper is clearly presented and relatively easy to follow. However, potentially avoiding large blocks of text and adding highlights or more visualisations may further improve it.

- **insight into existing theory**. The comparison with existing graph isomorphism framework on temporal graphs is useful, such as comparison with the timewise isomorphism recently proposed. My overall take is that there are many different possible notions of graph isomorphism on temporal graphs due to the extra degree of freedom introduced in the temporal dimension and having more diverse perspectives is useful.


- **included limitations.** The authors are also very frank and included a detailed limitations sections. Overall, I think this is useful for discussion and gaining insight into the authors' understanding of the current work and how it is positioned in the literature.

**Weaknesses:**

- **lack of TGNN baselines**. As the authors also pointed out, there is a lack of comparison to other TGNN architectures. This makes the empirical evaluation limited. While it is true that many TGNNs are designed for link prediction, recently there are also a number of papers focusing on graph tasks on temporal graphs. Here are some examples [1], [2], [3]. Comparing with SOTA graph task architectures on large real world datasets can better demonstrate the significance of the proposed architecture

- **Architecture categorisation**. While the proposed isomorphism is compared with current graph isomorphisms in temporal graphs, there is less detail on how current architectures would fall into the new consistent event graph isomorphism. Would some architectures be more powerful than 1-WL here? How to design more powerful architectures in this isomorphism notion?

- **Dataset statistics**. How big are the real world datasets used in RQ2? What are the dataset statistics? Can we link the task in RQ2 to some real world application?





[1] Shamsi K, Poursafaei F, Huang S, Ngo B, Coskunuzer B, Akcora CG. GraphPulse: Topological representations for temporal graph property prediction. InThe Twelfth International Conference on Learning Representations 2024 Mar. ICLR.

[2] Uddin MJ, Changani S, Coskunuzer B. T3former: Temporal Graph Classification with Topological Machine Learning. arXiv preprint arXiv:2510.13789. 2025 Oct 15.

[3] Pritam S, Roy R, Sajeev MC. Classification of Temporal Graphs using Persistent Homology. arXiv preprint arXiv:2502.10076. 2025 Feb 14.

**Questions:**

- while the standard link prediction and node classification tasks are not compared, do you think the proposed architecture would have benefits in these tasks due to the increased expressiveness? If so, in what ways?
- with recent scalable libraries for TGNNs such as [TGLite](https://github.com/ADAPT-uiuc/tglite), [TGM](https://github.com/tgm-team/tgm) and [TGL](https://github.com/amazon-science/tgl), do you think it would be possible to include more TGNN baseline results, potentially for more tasks as well?
- Typo on line 832, the reference Morris et al. is shown twice, similar to Rossi et al. on line 833

**Details Of Ethics Concerns:**

No ethical concerns

---

### Official Review · Reviewer_ejn6 · 2025-10-31

**Soundness:** 1
**Presentation:** 2
**Contribution:** 1
**Rating:** 2
**Confidence:** 4

**Summary:**

This work has attempted to define isomorphism for temporal graphs.

**Strengths:**

I appreciate the theoretical investigation of temporal graph isomorphism by the authors.

**Weaknesses:**

1. The term causality is used loosely throughout the paper.
2. Relevant prior art is not cited. For example, the idea of counterfactual testing through shuffled timestamps for temporal interaction graphs was introduced in `[R1]`. The authors have cited Pritam et al. (available on arXiv since February 2025), while the full version of `[R1]` is available online on [OpenReview](https://openreview.net/forum?id=k3LAIS5wTY) since October 2024
3. The causal model underlying the temporal interaction graphs is not presented.
4. The confidence intervals in the numerical results are not interpreted correctly.
5. On Line 852, the authors have mentioned *edge frequencies* but not cited `[R1]`, which introduced changing edge frequencies as a temporal distortion technique.

> [R1] Aniq Ur Rahman, Alexander Modell, Justin Coon Proceedings on "I Can't Believe It's Not Better: Challenges in Applied Deep Learning" at ICLR 2025 Workshops, PMLR 296:13-19, 2025.

**Questions:**

1. Where is the causal topology defined for the synthetic temporal graphs?
2. Checking isomorphism for graphs with labelled nodes is not a big issue; could the authors please clarify what the motivation is behind this work?
3. Occurrence of edge events in succession does not guarantee causality. Could the authors please comment on this statement?
4. What does bolding in Table 1 signify? If it means the best result, then I believe the confidence intervals are interpreted incorrectly. If the confidence intervals of two models overlap, one cannot claim that one model is better than the other.
5. Could the authors please comment on the performance improvement with respect to the dataset size and number of training samples?
6. Since the causality of real-world datasets is not established, what is the purpose of conducting experiments on them?
7. What is the rationale behind only choosing temporal graph classification for evaluation?

---

### Official Review · Reviewer_qiaV · 2025-11-01

**Soundness:** 3
**Presentation:** 3
**Contribution:** 3
**Rating:** 4
**Confidence:** 3

**Summary:**

The paper begins with theoretical considerations that motivate a GNN method for temporal graph classification. Theoretically, the paper proposes "consistent event graph isomorphism" as a middle-ground of strictness between two natural definitions of temporal graph isomorphism based on isomorphism of temporal paths, and shows that this middle-ground isomorphism is equivalent to standard (directed) graph isomorphism on an "augmented event graph" derived from the temporal graph. This motivates applying a prior "Dir-GNN" method for directed graphs to this augmented event graph, since the method has been proved as strong as the D-WL test, a directed version of the Weisfeiler-Leman test. This proposed method is then shown to have strong performance at temporal graph classification on proposed synthetic data, as well as 5 real-world datasets, on which it is compared to prior methods GAT and TGN.

**Strengths:**

- The paper addresses an interesting problem of temporal graph classification, which appears to have limited past exploration.
- The paper is generally clearly written and logically organized.
- The paper has a satisfying structure of theoretically motivating the proposed method.
- Regarding reproducibility, the experiment setup appears to be described in good detail in Appendix E. It is stated that code would be provided upon acceptance.

**Weaknesses:**

- The diversity of datasets is limited. This may be because the paper only focuses on temporal graph classification, and does not include node classification or edge prediction/classification, as the authors note. The paper would be strengthened if the authors either expand their focus, or provide more examples applications of temporal graph classification to justify its importance.
- Further, on datasets, the datasets don't seem to be described in either the main paper of the appendix. What is the node/edge count, what do the graph and labels represent, etc.?
- As the authors note, only comparing to two other methods is not ideal. The authors note the recency-bias and link prediction focus of prior works CAW/PINT as a reason not to include them, but I think a good-faith effort to adapt them to the setting (e.g., ablating the exponential time decay in PINT if it improves performance) would strengthen the experimental results.
- The main theoretical contribution of the work is Theorem 1-2. The proofs are appendicized, which is fine, but there should be some discussion of the proof strategy and techniques.

Typos / Minor
- Please use \citep for parenthesized citations where appropriate, e.g., at "temporal graphs Longa et al. (2023)" on the first page.
- On line 182, is the notation correct? As written, it's saying that an interaction between two entities happening at a given time doesn't necessitate another interaction between the two entities at another time. This could make sense, but maybe you could add that it also doesn't imply $(u, v; t') \notin E^{\tau}$.
- What is a "unit edge traversal time"? Is this the $1 \leq \Delta t$ constraint in Definition 3?

**Questions:**

- Is this the first work applying line graphs to temporal graph learning / TGNNs?
- The proposed method strongly outperforms TGN on the 3 other datasets, but the performance on the "ants" datasets is competitive or even favors TGN. Is there something about this dataset that reduces the performance gap?
- Assuming my understanding of "unit edge traversal time" above is correct, how do you adapt it (and the max time allowed $\delta$ between edges in a temporal path) to the real-world datasets?

---

### Official Review · Reviewer_knrP · 2025-11-06

**Soundness:** 2
**Presentation:** 2
**Contribution:** 1
**Rating:** 2
**Confidence:** 5

**Summary:**

Authors propose a notion of isomorphism for temporal graphs, show it can be reduced to isomorphism in directed static graphs, and use this relationship to propose a message-passing algorithm.

**Strengths:**

* Overall well-written and easy to follow

* Addresses a relevant theoretical problem in GNNs, i.e., the analysis of temporal graph models.

**Weaknesses:**

Below, I list my most pressing concerns with this work, which I believe would require major modifications to the manuscript.

* Limited technical novelty. Isomorphism on temporal graphs has been previously studied in the GNN literature and the technical developments (c.f., proofs) in this work appear fairly straightforward.

* The manuscript provides a weak and sometimes imprecise account of prior literature. In the abstract, authors state “we lack a generalization of graph isomorphism to temporal graphs that fully captures their causal topology”. To properly back their claims, I expect an in-depth discussion on how their definition — and the expressive power of their proposed message-passing algorithm — compare to those in Gao & Ribeiro (2022) and Souza et al. (2022). Also, the account of the latter reference around Line 154 is incorrect. Souza et al. (2022) show that CAW and TGNs are not equally expressive — see Fig 1 in their manuscript. In summary, I believe the positioning of this work to the prior literature ought to be clearly elucidated.

* Weak motivation. As I understand, when the authors refer to ‘causal influence’ they are broadly concerned with the flow of information between nodes respecting the order in which events occur. However, they often resort to vague terms like “causal topology”  to motivate their work. Despite having no clear definition, it is not clear how this relates to what the ML community understands as causality/causal inference.

* Weak experimental campaign. Authors only compare their method against TGN and a static base using datasets that do not match the recent literature in TGNNs. There are also no experiments in link prediction nor comparison against methods like PINT and CAW.

**Questions:**

See weaknesses above

---

### Note · Authors · 2025-12-02

I have read and agree with the venue's withdrawal policy on behalf of myself and my co-authors.